# Gut Microbiota in Nutrition and Health with a Special Focus on Specific Bacterial Clusters

**DOI:** 10.3390/cells11193091

**Published:** 2022-09-30

**Authors:** Lucas R. F. Bresser, Marcus C. de Goffau, Evgeni Levin, Max Nieuwdorp

**Affiliations:** 1Department of Internal and Vascular Medicine, Amsterdam University Medical Center, 1105 AZ Amsterdam, The Netherlands; 2HorAIzon B.V., 2645 LT Delfgauw, The Netherlands

**Keywords:** omics, microbiome, human health, diet pattern, machine learning

## Abstract

Health is influenced by how the gut microbiome develops as a result of external and internal factors, such as nutrition, the environment, medication use, age, sex, and genetics. Alpha and beta diversity metrics and (enterotype) clustering methods are commonly employed to perform population studies and to analyse the effects of various treatments, yet, with the continuous development of (new) sequencing technologies, and as various omics fields as a result become more accessible for investigation, increasingly sophisticated methodologies are needed and indeed being developed in order to disentangle the complex ways in which the gut microbiome and health are intertwined. Diseases of affluence, such as type 2 diabetes (T2D) and cardiovascular diseases (CVD), are commonly linked to species associated with the *Bacteroides* enterotype(s) and a decline of various (beneficial) complex microbial trophic networks, which are in turn linked to the aforementioned factors. In this review, we (1) explore the effects that some of the most common internal and external factors have on the gut microbiome composition and how these in turn relate to T2D and CVD, and (2) discuss research opportunities enabled by and the limitations of some of the latest technical developments in the microbiome sector, including the use of artificial intelligence (AI), strain tracking, and peak to trough ratios.

## 1. Introduction

One’s health is affected by environmental factors, diet preferences, social-economic status, and lifestyle [1,2,3], yet ample progress can still be made on how these factors and health are intertwined with the gut microbiome. The gut microbiome is highly diverse (alpha diversity), including many microbial species that have yet to be even cultured, and has a high degree of variability when comparing the gut microbiota compositions of people (beta diversity), even when living in a community, but especially when comparing people of different ethnicities living in different parts of the world [4]. Archaea, bacteria, viruses, and fungi are members of the gut microbiota [5], and all of these interact with one another and the host. These interactions are essential, or even fundamental, processes for host development, including endocrine signalling development, immune responses, and bile acid signalling and metabolism [6]. With the arrival of metagenomic and metabolomic techniques, it became feasible to examine gut compositions in various contexts (e.g., healthy/diseased) [7]. A “healthy” gut microbiome cannot, however, be generalized over a whole population, since every individual has a unique microbiome due to external and internal factors, such as environmental factors, genetics, and diet [4,6,8,9,10]. However, as shown in Figure 1, some (combinations of) certain species are repeatedly discovered to be related to or associated with specific diseases, conditions, and/or regions of the world. A commonly used abstraction of gut microbiome compositions is the concept of enterotypes. which are used to facilitate comparisons both within and between studies. The main two enterotypes that are commonly recognized are the *Bacteroides* and the *Prevotella* enterotypes. These two genera tend to be mutually exclusive (antagonistic) and hence a gut microbiome composition consisting of species that are compatible with either *Bacteroides* or *Prevotella* occurs. Samples without a high abundance of either of these two genera also exist between these two states, typically having a high abundance of Firmicutes instead. While a sample is defined as being of a particular enterotype, and hence a supposed distinct state, one should keep in mind that the reality is that a continuum of states exists between supposed enterotypes, and that within this continuum/gradient of states, some are more stable than others [11] (Figure 1). The *Prevotella* enterotype, dominated by the *Prevotella* genus, but also enriched with various species that are in a complex trophic network with some *Prevotella* species [4], is commonly found in rural populations in Africa [12,13,14], South America [15], or South-East Asia [16], is associated with a non-industrialized dietary fibre-rich diet, and is more commonly found in vegetarians. On the opposite side of the microbial composition gradient, one finds two *Bacteroides* enterotypes, commonly found in North America and/or urbanized environments. The *Bacteroides* 1 enterotype represents a fermentatively functional microbiome composition, with adequate microbial density and gene diversity levels [17], characterized by high levels of *Bacteroides*/*Phocaeicola* and *Blautia*, yet the *Bacteroides 2* enterotype constitutes a composition which should be considered dysbiotic. The *Bacteroides 2* enterotype is characterized by low microbial (gene) diversity, low microbial density, increased *Enterobacteriaceae* levels (amongst others), decreased levels of various (presumed) beneficial commensals, higher water content levels, and more likely still being in the saccharolytic phase of fermentation, and has been linked to systemic inflammation [17] and a plethora of other undesirable associations, such as IBD [18,19], obesity [17,20,21], and T2D [22]. People in various European countries often share a gut microbiome pattern dominant in the *Bacteroides* enterotype with a Firmicutes-enriched composition. Such a pattern is considered gut dysbiosis, which may be prone to increased risk of diseases (Figure 1) [4,14]. 

## 2. Confounding Factors and Mediators Affecting the Gut Microbial Composition

Although many microbiome studies provide essential insights into how intertwined the gut microbiota and one’s health are, one should keep in mind that correlations or associations do not equal causality and that many confounders and mediators obfuscate matters further. How these confounders are addressed (e.g., by matching, correction, or multi-omics) ultimately affects whether the intended results are legitimate and/or if appropriate conclusions can be drawn. In this section, we (1) discuss several complicated multi-factorial confounders commonly encountered while analysing the microbiome, (2) illustrate how some of the most prevalent internal and external variables affect the composition of the gut microbiome, and (3) show how these factors connect to various diseases and/or conditions.

### 2.1. Diet

While being an internal part of the human body, the digestive system and its microbiome are directly exposed to various external factors. such as diet, medication use, lifestyle choices [2], and environmental factors [3]. The gut microbiome directly contributes to the primary purpose of the human digestive system through the biosynthesis of vitamins and amino acids, as well as the production of critical metabolic by-products [23]. These by-products include short-chain fatty acids (SCFAs), such as butyrate, a prominent energy source for intestinal epithelial cells [24]. Secondly, bile acids constitute another category of metabolites generated by microbes. Numerous studies have noted a connection between the gut microbiota, bile acids, and obesity [25,26] Interestingly, species associated with a trophic network associated with the Firmicutes enterotype (e.g., *Christensenellaceae* and *Methanobrevibacter*) are involved in primary bile acid deconjugation via bile salt hydrolases [27]. 

High consumption of red meat, sugars, and fried foods at the expense of the intake of whole grains, fruits, fish, seeds, and nuts is typical in Western diets. As a result, Western diets contain high amounts of trans fats, saturated fats, and animal protein, and a low amount of dietary fibre [28]. Sugars, trans fats, saturated fats, and animal proteins are characterized as negative contributors to vascular health, since they are associated with the upregulation of blood cholesterol (total and LDL) [29,30,31,32]. Increases in these same dietary components are similarly associated with shifts in the gut microbiome towards compositions that are *Bacteroides*-dominated (*Bacteroides* enterotype) [33,34,35,36,37], especially the *Bacteroides 2* enterotype, which has a low microbial alpha diversity and typically increased *Enterobacteriaceae* levels [17,38], and is commonly linked to diabetes [22], obesity [17,20,21], and various other risk factors directly associated with cardiovascular disease (CVD) [39,40]. In the study by Lassen and Attaye et al. [37], it was illustrated using two cohort studies, HELIUS and MetaCardis [41,42], that taxonomical signatures could be associated with animal protein intake, regardless of ethnicity.

Although no current therapy offers a cure to obesity, lifestyle interventions with or without antidiabetic agents have been used to treat obese patients. Typically, lifestyle interventions focus on reducing caloric intake and increasing physical activity by incorporating behavioural modification [40,43]. The Mediterranean diet, rich in dietary fibre, fresh fruits, nuts, and vegetables [43], is commonly implemented as part of such interventions. Various (dietary) intervention studies are initially successful at inducing weight loss [40,44,45], yet they prove difficult to maintain in real-world settings, not uncommonly ultimately leading to a gain in weight shortly after the last intervention. This same pattern is mirrored by shifts seen in the gut microbiome of overweight subjects participating in such studies that are either on the Mediterranean [46] or other high-fibre-content diets [47,48]. In all high-fibre-content diet studies, a positive association between high fibre consumption and the abundance of species and/or genera related to the *Prevotella* enterotype has been observed. In the study by Hjorth et al. [47], 62 overweight subjects were randomly assigned to a whole grain diet or a regular diet, with the former resulting in greater weight loss rates. Of interest, however, was that patients that initially already had a high *Prevotella/Bacteroides* ratio (P/B ratio) lost more weight on average when on a whole-fibre diet than those with an initially low P/B ratio. In line with these findings, Kovatcheva-Datchary et al. [48] illustrated that improved glucose metabolism as a result of a high-fibre-content barley-kernel-based bread diet was coupled with an increase in the P/B ratio, and more specifically with the enrichment of *Prevotella copri*. In contrast, however, the Mediterranean diet has been reported to be potentially more helpful in ameliorating insulin sensitivity in subjects with a low P/B ratio [46]. Instead of increasing the *Prevotella* levels, the Mediterranean diet was found to increase the abundance of butyrate-producing species, such as *Faecalibacterium prausnitzii* and *Roseburia*, while resulting in a decrease in *Ruminococcus gnavus*, a species more commonly associated with the more dysbiotic *Bacteroides 2* enterotype [49]. Mirroring the high fibre–*Prevotella* connection and mimicking the effects of the Mediterranean diet, capsaicin supplementation has been found to mainly generate favourable effects in *Bacteroides* enterotype subjects and not in those of the *Prevotella* enterotype, and induce a shift towards more *Faecalibacterium prausnitzii* [50,51]. Similarly, a balanced, more traditional Korean diet, richer in capsaicin and fibre than a more Western diet, has been associated with a shift away from the *Bacteroides* enterotype(s) [49]. The effects of interventions or diet in general and the concomitant microbial shifts and metabolic changes hence should not be expected to generate uniform outcomes, but are dependent on the interplay between the type of diet/intervention and the initial microbial composition. *Prevotella* has, in general, been regarded as a beneficial contributor to health, as it is associated with a plant- and fibre-rich diet. However, some research indicates that *Prevotella* may also have a downside, as it is, for example, also associated with elevated serum uric acid levels, which are in turn associated with kidney-related disorders (e.g., kidney stones) [52]. Having a *Prevotella*-dominant microbiome combined (suddenly, in the case of immigration) with high sugar consumption [53] may lead to such complications.

### 2.2. Environment, Ethnicity, and Genetics

In the last few centuries, many parts of the world have seen rapid societal and concomitant dietary changes as a result of rapid industrialization and increased urbanization, which now seem to have reached a tipping point in the past few decades with regard to their impact on the gut microbiome, resulting in continuous increases in the obesity, T2D, and CVD rates [1]. Obesity is typically associated with an energy imbalance between consumed and burned calories, largely resulting from exercise choices and personal diet preferences. Increasingly, however, this imbalance commonly results from social and economic changes at levels beyond the control of individuals [54,55]. These so-called obesogenic changes include nutrient-poor food becoming less expensive, with the reverse pattern occurring for more nutritiously less-commendable foods, increased levels of mechanized transportation, and more sedentary lifestyles/jobs. 

The most common microbial compositions within a population depend largely on dietary culture and living conditions. Ethnicity is commonly strongly linked to the above two factors, yet when people migrate from one place to another, (dietary) acculturation often occurs, resulting in the gut microbiome of a particular ethnic minority becoming more similar to that of its host’s population over time, especially when viewed across generations [16]. When comparing people of the same ethnicity, it is hence important to take their history and habits into account. Acculturation, and as a result microbiome shifts, are, as said, far from instantaneous, as Dutch, Ghanaians, Moroccans, Turks, African Surinamese, and South Asian Surinamese living (for a long time) in Amsterdam are still quite distinguishable from one another from a gut microbiome perspective on a population/ethnicity-wide level. Despite frequently not yet having a gut microbiome composition similar to the host population (more *Bacteroides*-rich), the rates of diseases of affluence are commonly still higher in ethnic minorities in Western countries. This apparent contradiction is due to the fact that diseases of affluence are multifactorial. Immigrants commonly start at the bottom of the socioeconomic ladder, and there is a very strong correlation between income and the rate of diseases of affluence [55]; affluence in this respect should be regarded as a misnomer. Ghanaians, Moroccans, and Turks, despite having higher *Prevotella* abundances than Dutch people on average, all have higher rates of diabetes (~5% vs. 10 + % in the HELIUS cohort). Surinamese, and especially South Asian Surinamese, who have low *Prevotella* abundances and even higher *Bacteroides* abundances than the Dutch, on the other hand, have extremely high diabetes rates (one in four in the HELIUS cohort), highlighting this cumulative multifactorial effect [56]. 

Nevertheless, urbanization has a profound effect on the gut microbiome composition, typically causing a shift towards more *Bacteroides*-rich compositions with a lower alpha diversity [16,33,57,58]. The study by Keohane et al. [59] on Irish Travellers is particularly revealing with regard to the effects of a change in lifestyle. In this study, 118 adult Irish Travellers, of whom 87% were nomadic in childhood, were compared with non-industrialized and industrialized populations. These formerly nomadic individuals were forced to abandon their nomadic lifestyle due to legislation in 2002 and became (more) urbanized as they had to live in so-called ‘halting sites’ and state-sponsored housing. The gut microbiome composition of nomadic Travellers was, on average, initially in between those of the non-industrialized and industrialized populations, erring towards those of non-industrialized populations, but rapidly became more similar to that of the non-Traveller Irish urbanized population after legislation was implemented. This may be explained by the fact that the urbanized population has less interaction with cattle and other animals [60] and experiences higher levels of air pollution linked to the *Bacteroides 2* enterotypes [61].

While previous (Caucasian) GWAS studies suggested that genetics play a small role in the gut microbiota composition [62], more recent data from a more heterogenous (multiethnic) population cohort suggest that the gut microbiota diversity and composition are influenced by specific genetic single-nucleotide polymorphisms (SNPs) [61]. However, the largest genetic difference found between humans (one entire chromosome) affecting the gut microbiome is ethnicity-independent. Constipation more commonly occurs in females and their bowel movements tend to be slower [63], resulting, on average, in lower Bristol stool scale scores [64] and accompanying gut microbiome compositions. Indeed, men are far more commonly found to be of the *Prevotella* enterotype [42,65,66] in areas where not nearly everybody is either of the *Prevotella* enterotype (rural Africa) or of a *Bacteroides* enterotype (America). Ethnicity-associated gut composition-relevant genetic differences do, of course, exist, such as the higher prevalence of lactose intolerance in various Asian ethnicities. A lack of the mutation that confers lactose tolerance (rs4988235) leads to relatively decreased ability to absorb lactose into adulthood from the gut, leading to higher levels of available lactose in the large intestine [67] and resulting in higher *Bifidobacterium* levels. In Han Chinese people, lactose malabsorption is very common (>90%, [68]) and, despite the absence of animal milk use in traditional Chinese diets, bifidobacterial numbers remain relatively high, commonly resulting in *Bacteroides*–*Bifidobacterium*-rich compositions [69,70]. This, however, strongly contrasts with (culturally pastoral) Mongolians, who, despite also being, genetically speaking, lactose intolerant (>90%), obtain up to a third of their calories from milk products. Mongolians nonetheless rarely experience the expected levels of lactose-induced discomfort, possibly due to their *Prevotella*–*Bifidobacterium*-rich gut microbiome compositions [71], which affects the final balance of fermentation products.

The importance of various environmental factors with regard to the development of the gut microbiome is still uncertain. For example, it was assumed for a long time that the majority of the microbes originating from the outside world via food and/or saliva would be eliminated by the high-acidity environment of the stomach [72,73]. Additionally, it was believed that individuals with certain diseases or conditions, such as rheumatoid arthritis [74], inflammatory bowel disease [75], and colorectal cancer [76,77], had an increased ability to transport bacteria that are typically present in the mouth to the stomach. However, research by Schmidt and Hayward et al. [78] offered evidence that bacteria with an oral origin travel to the gut more frequently than previously believed. Increased oral hygiene has shifted oral microbiomes worldwide, causing a shift away from compositions rich in *Methanobrevibacter* [59,79], a species that also appears to be on the decline in the gut in more urbanized settings [59]. The colonization process of the gut with other orally derived species might be similarly affected.

### 2.3. Antibiotics Use and Other Drugs

Antibiotics are indispensable in the medical world, and their use continues to rapidly increase worldwide [80]. Concomitant with the increase in antibiotics, the levels of antibiotics resistance continue to increase, even in developed countries where they try to limit their use [81,82,83], posing a major threat to human health as some infections become untreatable due to strains becoming multi-resistant [84,85]. It needs no explanation that antibiotics disrupt the gut microbiome severely, possibly having long-term effects on health by disrupting/delaying normal gut microbiome development in infants [86,87] and in general by causing the disappearance of various strains that cannot re-establish themselves after treatment [88,89,90,91,92]. 

With the disappearance of one or multiple species within the gut ecosystem, trophic networks are especially vulnerable to antibiotics and other drugs, resulting in a lower alpha diversity [93] and a selection for compositions that are dominated by strains that are less co-dependent, such as *Prevotella copri* [94] or the *Bacteroides 2* enterotype. Both in the short and long term, many studies report that antibiotics tend to favour the *Bacteroides* enterotypes. The study by Palleja et al. [95] illustrated that, after a mixture of three so-called last-resort antibiotics (meropenem, gentamicin, and vancomycin), a bloom of various pathobionts occurred initially, followed by the recovery of the alpha diversity in the weeks thereafter, yet various important commensals, such as specific bifidobacterial strains, certain butyrate, producers and *Methanobrevibacter smithii*, a key member of trophic networks associated with various health benefits [96,97], never returned. Similar findings of (similar) species disappearing or being reduced in number are frequently reported [98,99,100,101]. Indeed, an increase in the Bacteroidetes phylum is often reported, which includes *Bacteroides*(/*Phocaeicola*) and *Prevotella copri* [98,99].

Owing to the indirect selection for the *Bacteroides* enterotype(s) by antibiotics, and owing to the link of this enterotype to various diseases, evidence is accumulating that antibiotics use may instigate/propel the development of such diseases. Recently a case–control study by Nguyen et al. highlighted that 23,982 individuals with IBD more frequently used broad-spectrum antibiotics than 117,827 healthy controls [102]. Prior to this, the study by Vich Vila et al. illustrated that individuals who suffered from microbial dysbiosis (e.g., Crohn’s) tended to have a higher level of the antibiotic resistance gene *cepA*, which is in turn correlated with the abundance of the *Bacteroides* genus [103]. Along with *cepA*, resistance genes such as *cfxA* and *cfiA* have been similarly linked to various *Bacteroides* strains [104,105,106]. The increase in the antibiotic resistance levels within *Bacteroides* spp. is also reflected by a decreased richness of *Bacteroides* strains [107], as more resistant strains naturally quickly outcompete non-resistant strains, even after antibiotics exposure; non-resistant strains may go extinct, and exclusion competition with resident surviving resistant *Bacteroides* strains may diminish their re-colonization chances.

Other medications, such as antidiabetics (metformin), statins, and beta-blockers, similarly affect the gut microbiome in various directions. In T2D diabetics, Forslund et al. [108] found that metformin induced a shift away from the Bacteroides1 enterotype towards the *Bacteroides 2* enterotype, whilst statins induced a shift towards the Firmicutes enterotype away from *Bacteroides 2*. Together, however, with the use of antibiotics, this led to an overall shift towards the *Bacteroides 2* enterotype, depletions of the Firmicutes and Prevotella enterotypes, lower microbial diversity, and a higher abundance of antibiotics resistance genes. Even before T1D is diagnosed (pre-diabetes), the gut microbiome is typically affected (possibly leading to T1D) as trophic networks supporting and including the butyrate-producing *Faecalibacterium* genus tend to be less abundant, while *Bacteroides* instead commonly seems to be elevated [109]. At the onset of T1D, this situation only seems to be even more aggravated, with not only even higher *Bacteroides* numbers, at the cost of *Prevotella* in populations where Prevotella is common [110], but frequently also with elevated *Enterobacteriaceae* numbers, possibly as the result of increased blood glucose levels. Sugar consumption is, by itself, also a risk factor for developing T1D [111] and also increases (inflammatory) *Enterobacteriaceae* numbers [112]. Gut microbiome compositions, however, typically normalize to a degree once diabetes is controlled [110]. As can be seen from the above, the situation is commonly a chicken and egg situation of factors aggravating the situation in tandem in a downward spiral to the bottom (*Bacteroides 2* enterotype).

### 2.4. Age 

Assuming a healthy start in life, all humans begin with a *Bifidobacterium*-dominated composition, yet once weaning commences, various complex adult-like gut microbiome configurations establish themselves rapidly within the first three years of life [12,113,114,115,116]. Various factors, however, such as antibiotics use, formula feeding, and the mode of delivery (Caesarean section), may delay or even prevent this initial *Bifidobacterium*-dominated composition, leading to diminished growth and immune-related issues years later in life [117,118]. Caesarean section has been found to be associated with an underrepresentation of *Bifidobacterium* and *Bacteroides* several months after birth and an increase in *Enterobacteriaceae* [119,120,121]. Although milk formulae have been improved to resemble breastmilk more closely, several studies still report that formula-fed infants have higher abundances of *Bacteroides* and *Clostridium* spp. compared with breastfed infants [122,123,124]. Furthermore, as new-borns move from a milk-based to a carbohydrate solid-foods-based diet, the abundance of *Bifidobacterium* decreases drastically and bacterial diversity increases concomitantly as a variety of species take its place [119,120]. Depending on the dietary components (e.g., animal proteins, fats, and high fibre) and environmental factors [125,126,127], the gut microbiome will develop in a particular direction, reaching a more “adult-like” state within three to five years after birth [15,128]. Beyond this, alpha diversity will continue to increase, albeit at a much lower rate, and enter a stable phase after the age of twelve [6]. Population-wide, it can be said that the alpha-diversity again starts to decline with age after 70 [6,129], yet in elderly people reaching extremely high ages, such a decline is not observed. One study by Biagi et al. [130] showed that Italian people over the age of 100 years still had high levels of bacterial groups associated with health, such as *Christensenellaceae* (associated with *Methanobrevibacter*), *Bifidobacterium*, and *Akkermansia*. Comparable trends with the same indicator species were also seen in Chinese individuals who lived long lives [129,131], indicating that a connection exists between the development of one’s gut microbiota and one’s health over one’s entire lifespan. Not only is the trophic network of which *Christensenellaceae* and *Methanobrevibacter* are key members associated with extremely high age, it is also linked to low blood pressure [132], reduced body weight [133,134,135,136], and reduced blood triglyceride values [17,135,136,137]. 

## 3. The Latest Technical Development in the Microbiome Field

### 3.1. The Rise of Machine Learning in Gut Microbiota Analyses

Artificial intelligence (AI) is becoming indispensable in today’s world. From cars to mobile (smart) phones, AI can help interpret patterns in the real world [138] and the same is true in the medical world, and, more specifically, in the gut microbiome field. Species within the gut microbiome interact with the host and one another, but these interactions often tend to be non-linear. Most machine learning methodologies allow one to capture these non-linear patterns. To achieve good performance, there is always a trade-off between the model’s complexity (e.g., number of variables, trees, or layers) and the number of samples. For gut microbiome intervention studies, the number of subjects varies between double and quadruple digits [56,139,140,141,142,143]. In addition, the model performance depends heavily on the phenotype of interest. For machine learning or AI concepts, the number of samples in microbiome studies tends to be on the low side [144]. For these types of studies where only relatively few subjects are included, decision-tree-based models, such as xgboost [145], gradient boosting [138,146], and extreme random decision trees [147,148], are often applied since these models are known to perform well when dealing with many variables compared with the number of subjects [129,140,141,142,143]. Although research questions differ across different studies, the type of information obtained from these methodologies can be generalized, i.e., which biomarkers or features are involved in distinguishing the target of interest. Along with the implementation of improved sequencing techniques, more unique, yet uncultured, species/sequences are identified as features of interest by these machine learning models. 

For example, research by Thomas et al. [141] showed that decision trees are capable of identifying biomarkers via feature importance that appear to be risk factors with regard to colorectal carcinoma development. These include *Fusobacterium nucleatum*, *parvimonas* spp., and *Solobacterium moorei*, species that were also suggested to be important in earlier colon cancer studies [149,150]. Contrary to expectations, patients with carcinomas have been shown to have higher bacterial alpha diversity when compared with healthy individuals. The authors hypothesize that this increased gut microbiome diversity might be due to them also harbouring a comparatively large number of orally derived species. 

Similarly, another study by Balvers et al. [140] showed that T2D South Asian and African Surinamese participants could be identified using a decision-tree-based machine learning approach. The authors further demonstrated how machine learning may be used to identify novel T2D biomarkers that have not previously been identified using traditional statistics. Here, bacterial strains connected to the species *Anaerostipes hadrus*, a butyrate producer, were associated with health. Despite being a butyrate producer, a characteristic commonly, but not always, associated with health, *Anaerostipes hadrus* has been found to be strongly associated with *Bacteroides enterotypes*. Most Surinamese people living in Amsterdam, especially those of South Asian origin, are, however, of the *Bacteroides* enterotype. Within this enterotype, quite a lot of variation is possible and *Anaerostipes hadrus* is indeed likely a good indicator of health, as it can be said to be on top of the food pyramid of metabolic products, producing butyrate as an end product, indicative of an adequate fermentative capacity [140]. In addition, using multivariate analysis for both ethnic groups’ usage of metformin appears to follow a similar pattern in terms of a strong correlation with *Escherichia/shigella*, which is known to be connected to the *Bacteroides 2* enterotype, thereby confirming the findings of Forslund et al. [108] that metformin use plays a confounding or mediating function in the development of the *Bacteroides 2* enterotype gut microbiome profile. Interestingly, a possibly ethnic-driven confounding effect can be observed when comparing both ethnic groups. Correlations in South Asian Surinamese with metformin use were stronger for a variety of species, including *Faecalibacterium* spp., *Blautia* spp., and *Anaerostipes*. Furthermore, significantly lower alpha diversity was only found in South Asian Surinamese with type 2 diabetes who used metformin compared with diabetics who did not. It is important to note that the alpha diversity in African Surinamese people is higher than that in Asian Surinamese people, as the latter often lack *Christensenellaceae*, *Methanobrevibacter*, and species associated with these groups. Similar effects of ethnicity and medication use have been observed in pathway analyses, including large negative associations within the isoleucine and lysine pathways and a minor positive association within the pathways for menaquinone and demethylmenaquinol production. This indicates that, while machine learning models can offer innovative targets and insights, connecting these findings to functionality or causality remains challenging due to various potential confounding factors. In order to better comprehend the revealed consequences of the exposed targets acquired by these complex non-linear machine learning models, one might use or explore cross-sectional processes across many different data domains (e.g., the microbiome, metabolomic, clinical, or epigenome) (Figure 2).

### 3.2. Multi-Omics 

Multi-omics approaches are increasingly being implemented to understand the associations and functionality of how microbiome biomarkers are linked to a target phenotype (e.g., disease or the outcome of an intervention), as well as with other data types, such as clinical, metabolomic, and epigenomic data [139,151,152,153]. These multi-omics analyses primarily include multivariate regression models combined with or without individually trained machine learning models. One study by Kootte et al. [139] demonstrated the effects of faecal microbiota transplantation (FMT) on the gut microbiome and on the blood plasma metabolite levels. Microbiome and metabolomic profiles were produced using a combination of traditional statistics, clustering methods, and rigid regression models revealing a significant influence of FMT on both bacterial diversity and insulin sensitivity. On the species level, a few lactate- and butyrate-producing species were found, including *Lactobacillus salivarius, Butyrivibrio, Clostridium symbiosum*, and *Eubacterium spp.* that distinguish allogeneic (faeces from donor) from autologous recipients (faeces from the self). Furthermore, within the allogeneic group, responders were characterized by an increase in *Akkermansia*, which is commonly associated with health [129,131]. Interestingly, no significant effects were reported in this study with regard to *Prevotella* and *Bacteroides*-related species in contrast to the follow-up study by van der Vossen et al. [154]. This study reported that modifications in the gut microbiota, specifically increased levels of *Prevotella* after an allogeneic FMT, might be connected to specific plasma metabolite levels and modifications in DNA methylation in plasma blood mononuclear cells by including epigenome data in addition to the data used in the study by Kootte et al. An increase in *Prevotella* was only observed in a minority of allogeneic FMT recipients, as most FMT donors also did not harbour substantial levels of *Prevotella*, yet its effect seemed most profound on the other data types, in comparison with the microbiome changes reported earlier by Kootte et al. These two studies highlight how more can be learned using more data types and incorporating improved methodologies.

However, combining different data sources remains a challenge, as certain states (health/disease) are commonly reflected across the different data domains. To combine the knowledge contained within the different domains, one might consider a manifold approach, yet such an approach might not be directly applicable to transfer the knowledge to one particular target domain. In one study by Pereira et al. [155], an approach based on manifold mixing combined with stacked regularization was developed to account for this problem. In their approach, a mixed manifold of the various data modalities was first created and used to merge one meta-model, including all of the different data types. By including a meta-model, more direct effects across different data domains, and thus across different parts or systems of the system, can be exposed. A follow-up study by Pereira and Bresser et al. [153] showed how additional data sources, such as faecal microbiome, clinical, nutrition, and plasma metabolomics data, when paired with a stacked regularization model, may achieve excellent model performance and uncover biomarkers associated with the impacts of the examined intervention. In spite of the fact that several cofounders were controlled throughout these studies, it is clear that mediators, such as genetics, diet, and sex, in the modality considerably aided in the interpretability of the data, and even potential gut–axial linkage associations across different domains. Numerous *Bifidobacterium* species were shown to be closely connected to the glutamate, lactic, and succinic acid blood levels, and of particular relevance are *Veillonella* and the relationship between the intestinal GABA levels known for being related to brain diseases [156,157].

### 3.3. The Underlying Linear Motor behind Microbiome Machine Learning Analysis 

The use of machine learning in the microbiological sector is growing along with new methodologies to reveal particularly novel species in addition to existing models. The enhancement of the interpretability of the model’s decision-making has received great attention during the creation of these new approaches. Decision trees use approaches based on feature importance to reveal which features or variables are linked to the performance of a model, which allows it to discriminate successfully between two or more groups. Although the feature importance of decision trees provides information about the creation of the model, it only provides indirect information about why the model made a certain decision [158]. Various approaches, which may be used with other machine learning and deep learning models, have been developed to address this issue, including permutated importance [148,158], pairwise permutated importance [159], and local interpretable model-agnostic explanations (LIME) [160]. These approaches assess the impact of all variables on the overall model and provide direct information about the effects of the variables. However, these non-linear decision tree-based models are more complex than traditional linear regression models. They are not complex compared with deep neural networks, and biologically relevant relationships/information might be left unexposed. Inspired by the imaging world, where neural networks are becoming the gold standard for segmentation and classification purposes for various imaging types (e.g., CT, MRI, and PET), including additional artificially generated data might improve the development of such deep neural networks for microbial-based predictive purposes, since a massive amount of data is required [144]. In the imaging world, generative adversarial networks (GAN) are developed to create artificial images from real data with generally more noise. Due to their excellent performance, GANs have been adopted in many fields, including biomedical research. One particular study by Rong et al. illustrated that, by applying Microbiome Generative Adversarial Networks (MB-GANs), the gut microbiome properties, including sparsity, diversities, and taxa–taxa correlations, remained intact when compared with real gut microbe data [161].

Even though significant advancements have been made over the years with regard to revealing novel cultured and uncultured species, methodologies for understanding the non-linear links between the species and features derived from other data domains are still underdeveloped. To this day, traditional statistical techniques, such as univariate tests, linear regression, and correlations, have mainly been used to highlight these relationships within species and between other areas [139,153,154]. Studying linear relationships and patterns, while comparing them with comparable studies, renders it more crucial and/or straightforward to make certain connections between what has been observed by the machine learning model and phenotype. However, one could question if these standard approaches can represent the outputs of machine learning from technical and biological viewpoints, as the non-linear biomarkers are reflected and compared with results derived from linear analyses. Laboratory investigations, such as mice or cell models, are often required to determine causal relations. It is, however, frequently impossible, from a practical point of view, to unravel all of these complicated relationships among the (many) relevant features through lab investigations. More biologically based approaches are, however, emerging and being used in the microbiome field to perform pathway analyses, including kinetic/ordinary differential equation (ODE) models [162,163] and the study of metabolic pathways based on energy generation and composition-based models [164,165].

### 3.4. Microbiome Viability and Origin 

Numerous analyses have been carried out with the presumption that the species discovered by sequencing are strains that were alive at the time of sampling. The fact that a particular microbial strain in such a high-microbial-biomass sample, such as faeces, can be sequenced led many to believe that this presupposition is true. For highly abundant strains, this is indeed likely the case, yet for many strains of intermediate abundance, this is not always clear. New technological advancements have made it feasible to ascertain the level of viability in terms of active duplication. Research by Korem, Zeevi, Suez, and Weinberger et al. [166] illustrated how the read coverage for various microbial genomes comprises a single peak and a single trough lead, which subsequently correspond to the bacterial source of replication. The ratio between the peak and trough provides a quantitative method of assessing the growth rate of a species or its viability. Although this method shows promise, it is currently difficult to use in microbiome studies for all species as complete reference genomes are required for all strains present within a sample and the sequencing coverage of a strain should be high enough [166,167]. A multisampling approach called dynamic estimator of microbial communities (DEMIC) was developed by Gao and Li [168] based on contigs and coverage values to determine growth rates using the relative distance from the origin of replication, allowing one to estimate the growth rates of still unknown or not fully sequenced species. Throughout the course of the investigation, reliable performance was attained across a range of sample sizes and assemblies. Despite the fact that these technical advancements show great promise, it is worth noting that they only allow for strain-level analysis at the species level. 

With regard to the origin of species themselves, strain-tracking techniques have been able to show links between the microbiomes of various body sites, such as the oral and the gut microbiome [78]. In the study by Schmidt and Hayward et al., associations or correlations were based on single-nucleotide variants (SNV) from which a transmission score was calculated. New methods are emerging that offer more conclusive evidence of the strain engraftment of the microbiome from the oral area into the gut. These methodologies incorporate shotgun-read metagenomics sequencing of SNV using probabilistic or non-probabilistic methods [169]. It has been shown in the past that probabilistic approaches are more advantageous in terms of sensitivity and precision [170]. The field is still evolving rapidly and methodologies with even greater performance are expected to become available in the upcoming years. Despite the fact that strain tracking is still in its infancy, investigations concentrating on FMT and gut microbiome intervention studies have already demonstrated its potential. As with the FMT trial by Kootte et al. [139], one study by Wilson et al. that used strain tracking on a lean donor FMT capsule intervention showed a significant beneficial shift in the direction of the enterotype similar to that of the corresponding donor. One study by Li et al. [171] further demonstrated that donor-specific microbiome strains were initially absent in the recipient, but could be detected two days after the intervention and were still found in these recipients three months later. 

### 3.5. Sequencing Limitations and Future Directions 

16S sequencing is still one of the most applied sequencing techniques in the microbiome field. Within the 16S ribosomal gene, one or combinations of multiple so-called variable regions (e.g., v1 and v3) are sequenced and subsequently analysed with various pipelines and reference libraries to ascertain the relative abundances of microbial groups [172,173]. Studies utilizing the same variable regions can be compared to a degree, if raw reads are processed through the same pipelines, yet differences in the DNA isolation protocol and primer choice, or even operator differences will still cause batch effects. As a result, only qualitative comparisons are typically feasible. Shotgun sequencing studies have become more popular over the years as they, amongst other advantages and capabilities, are not dependent on a specific variable region or on primer choice, yet comparisons between studies will still suffer from batch effects due to differences in the pre-sequencing processing steps [174]. Furthermore, while the relative abundance describes abundances within a single sample, comparing samples in the same manner remains challenging, as fractions do not equal the absolute bacterial abundances. One solution to this issue is to use a 2D method that considers both the absolute microbial load and the relative abundance of microbial groups, which can be achieved by cell counting techniques [175,176] or by including a positive control of a known quantity within every sample before performing DNA isolation [177]. A current disadvantage of shotgun sequencing, apart from its cost, is that its data analysis pipelines [178] are still reference library-dependent, while the latest 16S data analysis pipelines generate Amplicon Sequencing Variants (ASVs) independently of a reference library. Sequencing technologies continue to improve at an impressive pace, and this is particularly the case for Nanopore sequencing [179]. While its utility was limited in the past due to high error rates, it has now become a viable alternative to 16S and shotgun sequencing, as it can, for example, sequence the entire 16S gene at high sequencing depth and accuracy, enabling species-level identification, and has the advantage of speed and potentially even being of clinical diagnostic use at a relatively low cost, as samples are sequenced individually instead of in batches [180].

## 4. Conclusions

The gut microbiome continues to garner attention from the public at large as its role in human health becomes increasingly undeniable. It has been demonstrated that the gut microbiome coevolves with its human host and that this is an ongoing process as society is (again) changing rapidly. Although many consequences of internal and external factors remain to be further elucidated, certain patterns form a reoccurring theme. The formation of *Bacteroides* enterotypes is generally stimulated as a result of the high levels of sugar, animal protein, and fat consumption that come with Western diets. High-fibre diets, in contrast, typically have a positive contribution to one’s health. The effect of this type of diet, however, seems to depend on the initial gut microbiota composition of the individual, illustrating that the effect of diet is significant, but also that its multifactorial effects on health are modulated by a variety of internal and external factors. Along with diet, the increased use of antibiotics selects for less diverse and less co-dependant microbial compositions, typically of the *Bacteroides* enterotype, and *Bacteroides* strains that are resistant to these antibiotics. This, in turn, is linked to severe dysbiosis and obesity, and the accompanying risk factors, such as T2D and CVD. Although these associations were linked to the *Bacteroides* enterotype, one should consider that correlations or associations do not infer causality or functionality. Multi-omics analyses aid in the interpretation of the relationships between certain species’ functions or mechanisms and their impact on human health. By incorporating data from different data domains and locations, more can potentially be learned, as certain phenotypes, such as disease reverberate, are caused by factors in different data domains. In addition to more established approaches, such as diversity analyses, clustering/enterotypes, and regression, novel methods, such as AI, peak-to-trough, and strain tracking, are becoming increasingly popular for studying the microbiome. AI has exhibited significant promise for capturing non-linear relationships between various species and phenotypes. However, as the human mind is mainly capable of comparing variables on a linear scale, it often remains difficult to interpret AI findings, which confounders, such as diet, ethnicity, medication use, and the environment, already made challenging. By incorporating multi-omics data into one or multiple modalities, certain confounders can be included in these analyses. In addition, as sequencing techniques evolve in the coming years, more advanced strain-tracking techniques will become available to connect species across different origins of the human body. The combination of established and novel techniques and microbial ecological insight will be needed to answer challenging questions with regard to a variety of gut axes, such as the brain–gut and oral–gut axes and their interplay with our diet.

## Figures and Tables

**Figure 1 cells-11-03091-f001:**
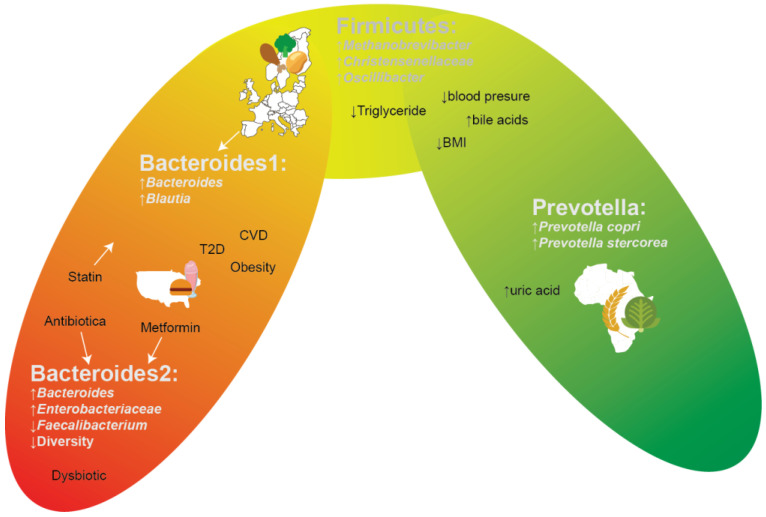
Gradient-like depiction of enterotypes and their corresponding associations.

**Figure 2 cells-11-03091-f002:**
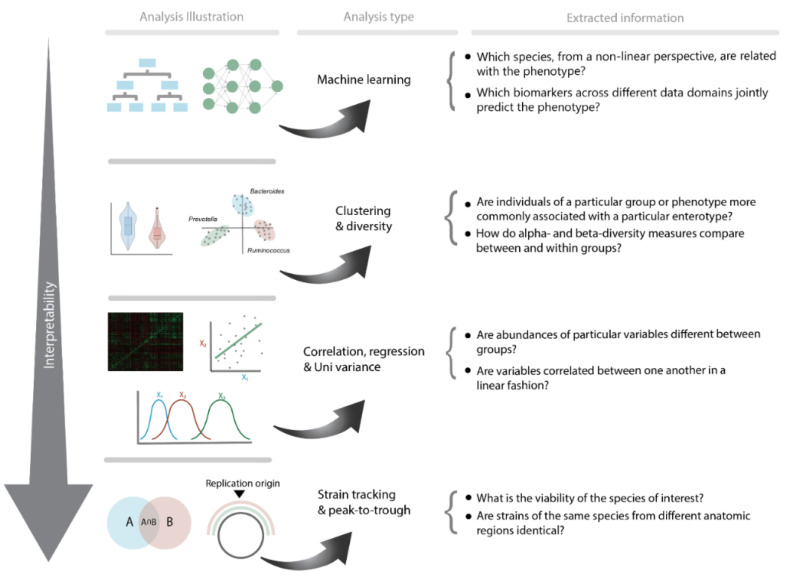
Overview of the most commonly used methodologies to examine microbiome data and the corresponding information accordingly.

## Data Availability

Not applicable.

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
