# Peer review of "Gut Microbiota in Nutrition and Health with a Special Focus on Specific Bacterial Clusters"

_cells, 2022, doi:10.3390/cells11193091_

Round 1

Reviewer 1 Report

Overall, the manuscript is interesting and useful for the field. To make it more readable for general audiences, I suggest revising the manuscript as follows.

1.       Line 46-51: A commonly used abstraction of gut microbiome compositions…. continuum/gradient of states some are more stable than others

The above statements are difficult to understand for broad-based scientists. Please consider delivering the message in a more simple way.

2.       Line 67-69: Various European countries can be sad to be more located between the Bacteroides and Prevotella enterotypes, though erring towards Bacteroides, with a more Firmicutes enriched composition

The above statements need revision as follows.

People in various European countries often share a pattern of the gut microbiome dominant in Bacteroides enterotype with Firmicutes enriched composition. Such a pattern is considered gut dysbiosis, which may prone to increased risk of diseases (Figure 1).

3.       Line 75-78: Although many microbiome studies provide essential insights into how intertwined the gut microbiota and one’s health are, one should keep in mind that correlations or associations do not equal causality and that many confounders and mediators obfuscate matters further.

Please add a comma as highlighted.

4.    Line: Various (dietary) intervention studies are initially successful at inducing weight loss [38, 42, 43] yet prove difficult to maintain in real-world settings not uncommonly ultimately leading to a gain in weight shortly after the last intervention.

The above statement is lengthy. Please break it into two sentences for better clarity.

5.       Line 169-170: Immigrants however commonly start at the bottom of the socioeconomic ladder which at least in part explains this observation for reasons described above [53].

Please elaborate more on the above statement. Why does starting from the bottom socioeconomic ladder explain the unchanged gut microbiome and high rates of disease in minorities?

6.    Line 349-351: Furthermore, a significantly lower alpha diversity was only found 349

in South Asian Surinamese with type 2 diabetes who use metformin compared to diabetics 350

who don’t.

Please revise the highlighted spoken language to a written one.

7.         Line 356-359: This illustrates that while machine learning models can offer innovative targets and insights it remains challenging to connect these findings to functionality or causality due to various potential confounding factors.

Please add some suggestions to improve the use of AI, after the sentence.

8.         Line 403-406: Numerous 403

Bifidobacterium species were shown to be closely connected with glutamate, lactic, and succinic acid blood levels, and of particular relevance are Veillonella and the relationship between intestinal GABA levels known for being related to brain diseases.

Please add references for the statements

9.    Line 444-447: However, one could question if these standard approaches can represent the outputs of machine learning from a technical and biological viewpoint as the non-linear biomarkers are reflected and compared to results derived from linear analyses.

Please add some suggestions to improve the analytical performance, after the sentence.

10.     Line 493-494: Although many consequences of internal and external factors remain to be further elucidated, certain patterns form a reoccurring theme

Please add a comma as highlighted.

11.     Line 497-500: The effect of this type of diet however seems to depend on the initial gut microbiota composition of the individual illustrating that the effect of diet is significant but that its multifactorial effects on health are modulated by a variety of internal and external factors.

The above statement is lengthy. Please break it into two sentences for better clarity.

Author Response

Overall, the manuscript is interesting and useful for the field. To make it more readable for general audiences, I suggest revising the manuscript as follows.

We thank the reviewer for her/his valuable comments, please find herewith our itemized response.

R1Q1.       Line 46-51: A commonly used abstraction of gut microbiome compositions…. continuum/gradient of states some are more stable than others

The above statements are difficult to understand for broad-based scientists. Please consider delivering the message in a more simple way.

R1A1: We’ve added couple of sentences to give a more basic understanding of why different enterotypes exist. Hopefully this will bridge the knowledge gap that some readers might have in regards to understanding what was stated in lines 46-51.

R1Q2.       Line 67-69: Various European countries can be sad to be more located between the Bacteroides and Prevotella enterotypes, though erring towards Bacteroides, with a more Firmicutes enriched composition

The above statements need revision as follows.

People in various European countries often share a pattern of the gut microbiome dominant in Bacteroides enterotype with Firmicutes enriched composition. Such a pattern is considered gut dysbiosis, which may prone to increased risk of diseases (Figure 1).

R1A2:
Thank you for the comment. We have changed the text accordingly.

R1Q3.       Line 75-78: Although many microbiome studies provide essential insights into how intertwined the gut microbiota and one’s health are, one should keep in mind that correlations or associations do not equal causality and that many confounders and mediators obfuscate matters further.

Please add a comma as highlighted.

R1A3:
Thank you for the comment we’ve added it accordingly.

R1Q4.    Line: Various (dietary) intervention studies are initially successful at inducing weight loss [38, 42, 43] yet prove difficult to maintain in real-world settings not uncommonly ultimately leading to a gain in weight shortly after the last intervention.

The above statement is lengthy. Please break it into two sentences for better clarity.

R1A4:
We thank the reviewer for the comment. We’ve split the statement into two parts.

R1Q5.       Line 169-170: Immigrants however commonly start at the bottom of the socioeconomic ladder which at least in part explains this observation for reasons described above [53]. Please elaborate more on the above statement. Why does starting from the bottom socioeconomic ladder explain the unchanged gut microbiome and high rates of disease in minorities?

R1A5:
We have made our point more clear by illustrating it using diabetes rates in the different ethnicities included in the HELIUS cohort whilst comparing these with their gut microbiome composition. Various ethnicities (Moroccan, Turks, Ghanaians) have higher Prevotella abundances (and lower Bacteroides abundances) than Dutch yet have higher Diabetes rates (~ 1 in 10) whilst Surinamese, and especially South Asian Surinamese have lower Prevotella abundances and higher Bacteroides abundances and having diabetes rates of ~1 in 4. Diabetes, as is generally known, is very multifactorial. living conditions and microbiome (and several other factors) are contributors.

R1Q6.    Line 349-351: Furthermore, a significantly lower alpha diversity was only found 349

in South Asian Surinamese with type 2 diabetes who use metformin compared to diabetics 350

who don’t.

Please revise the highlighted spoken language to a written one.

R1A6: 
We have changed this accordingly.

R1Q7.       Line 356-359: This illustrates that while machine learning models can offer innovative targets and insights it remains challenging to connect these findings to functionality or causality due to various potential confounding factors.

Please add some suggestions to improve the use of AI, after the sentence.

R1A7:
We thank the reviewer for the comment. We’ve added a suggestion that multi-omics strategies should be included in AI as well. We illustrated how this can be achieved in the section “multi-omics”.

R1Q8.       Line 403-406: Numerous 403 Bifidobacterium species were shown to be closely connected with glutamate, lactic, and succinic acid blood levels, and of particular relevance are Veillonella and the relationship between intestinal GABA levels known for being related to brain diseases.

Please add references for the statements.

R1A8:
An additional reference has been added.

R1Q9.    Line 444-447: However, one could question if these standard approaches can represent the outputs of machine learning from a technical and biological viewpoint as the non-linear biomarkers are reflected and compared to results derived from linear analyses.

Please add some suggestions to improve the analytical performance, after the sentence.

R1A9:

As suggested by the reviewer, some suggestions on how to improve the analytical performance have been added. Here, we’ve mainly focused on biologically explaining models such as kinetic models and Gibbs free energy-based models.

R1Q10.     Line 493-494: Although many consequences of internal and external factors remain to be further elucidated, certain patterns form a reoccurring theme

Please add a comma as highlighted.

R1A10:  
Done.

R1Q11.     Line 497-500: The effect of this type of diet however seems to depend on the initial gut microbiota composition of the individual illustrating that the effect of diet is significant but that its multifactorial effects on health are modulated by a variety of internal and external factors.

The above statement is lengthy. Please break it into two sentences for better clarity.

R1A11:  
We’ve split the statement above into two parts. Thank you for the suggestion.

Reviewer 2 Report

This review provides an overview of the role of gut microbiota in human health and also discusses specific bacterial clusters. Several previous studies described the role of the gut microbiota and its effects on human health are already well described, however, the authors have provided a clear and balanced view of what is known. I found the review easy to read, and from my literature searches, the review appears to be fairly comprehensive, and topical, even if so much has been published on this subject. However, some points need to be addressed to improve the quality of the paper.

1-Please also describe how? gut microbiota is altered with disease progression or Vice Versa.

2- Microbiota plays both positive and negative roles in human health, and also needs to be discussed briefly in the text.

3- In addition, the role of gut microbiota-derived secondary metabolites regulates human health needs to be discussed here.

4- Limitations and future directions in the gut microbiota analysis also need to be discussed.

5- The graphical illustrations need more extensive pieces of information (microbial metabolites and their role in diseases progression)

Author Response

This review provides an overview of the role of gut microbiota in human health and also discusses specific bacterial clusters. Several previous studies described the role of the gut microbiota and its effects on human health are already well described, however, the authors have provided a clear and balanced view of what is known. I found the review easy to read, and from my literature searches, the review appears to be fairly comprehensive, and topical, even if so much has been published on this subject. However, some points need to be addressed to improve the quality of the paper.

We thank the reviewer for her/his valuable comments, please find herewith our itemized response.

R2Q1-Please also describe how? gut microbiota is altered with disease progression or Vice Versa.

R2A1:
We appreciate the suggestion. We have provided an example concerning (pre-)diabetes versus treated T1D in section 2.3 where it is obvious that multi-factorial effects have a role in the development of the illness. Here, we emphasized that several variables have a role in both the establishment of the gut microbiome and the course of T1D.

R2Q2- Microbiota plays both positive and negative roles in human health, and also needs to be discussed briefly in the text.

R2A2: 
We thank the reviewer for the suggestion and comment. We have added extra microbiome studies that highlight the positive effect of species from a trophic network associated with the Firmicutes enterotype that are highly prevalent in healthy elderly (100<years), are associated with low blood pressure, and BMI (please see section 2.4). We would like to emphasize that correlation is not equal to causality, and therefore we only can speculate about both positive and negative roles in human health. For instance, whiles focusing on the Firmicutes to Bacteroidetes ratio conflicting relations regarding blood pressure and body weight have been shown. Suggesting that the effect or role of Firmicutes on health is heavily dependent on many factors.

R2Q3- In addition, the role of gut microbiota-derived secondary metabolites regulates human health needs to be discussed here.

R2A3:
In section 2.1, we've given more examples of secondary metabolites generated from the microbiota and how some of these compounds are connected to illnesses including obesity and the associated risk factors, kidney stones, and CVD.

R2Q4- Limitations and future directions in the gut microbiota analysis also need to be discussed.

R2A4:
We’ve added an extra section “Sequencing limitations and future directions” (section 3.5). Here, technical (sequencing) limitations and undesired sources of variance within and between different samples and batches are explained.

R2Q5- The graphical illustrations need more extensive pieces of information (microbial metabolites and their role in diseases progression)

R2A5:
We thank the reviewer for the suggestion and comment. Derived from 2th and 3rd comments additional metabolites and disease progression are added to the figure.

Round 2

Reviewer 2 Report

I am satisfied with this manuscript.